# A High-Robust Sensor Activity Control Algorithm for Wireless Sensor Networks

**DOI:** 10.3390/s22052020

**Published:** 2022-03-04

**Authors:** Rong-Guei Tsai, Xiaoyan Lv, Lin Shen, Pei-Hsuan Tsai

**Affiliations:** 1New Engineering Industry College, Putian University, Putian 351100, China; rgtsai@163.com (R.-G.T.); sig73ma@gmail.com (X.L.); shl8354@163.com (L.S.); 2Institute of Manufacturing Information and Systems, National Cheng Kung University, Tainan City 701, Taiwan

**Keywords:** quality of service, Gur Game, wireless sensor networks, internet of things, power saving

## Abstract

In wireless sensor networks, it is important to use the right number of sensors to optimize the network and consider the key design and cost. Due to the limited power of sensors, important issues include how to control the state of the sensor through an automatic control algorithm and how to power-save and efficiently distribute work. However, sensor nodes are usually deployed in dangerous or inaccessible locations. Therefore, it is difficult and impractical to supply power to sensors through humans. In this study, we propose a high reliability control algorithm with fast convergence and strong self-organization ability called the sensor activity control algorithm (SACA), which can efficiently control the number of sensors in the active state and extend their use time. In the next round, SACA considers the relationship between the total number of active sensors and the target value and determines the state of the sensor. The data transmission technology of random access is used between the sensor and the base station. Therefore, the sensor in the sleep state does not need to receive the feedback packet from the base station. The sensor can achieve true dormancy and power-saving effects. The experimental results show that SACA has fast convergence, strong self-organization capabilities, and power-saving advantages.

## 1. Introduction

### 1.1. Background

Wireless sensor networks (WSNs) are critical technologies of the Internet of Things, and they are mainly composed of base stations and hundreds or thousands of sensors. Because the power of sensors is limited, and sensors are usually deployed in dangerous or inaccessible locations, it is difficult and impractical to supply power to sensors in areas that are not easy for humans to reach. Therefore, controlling the state of the sensor through an automatic control algorithm and maximizing the function of the sensor, power saving, and efficient distribution of the mission have become important issues. Thus, efficiently controlling the number of sensors in the active state improves their working strategy and efficiency and extends their lifetime [1,2,3,4]. Moreover, the number of sensors in the active state should be controlled according to the application requirements. For example, the number of sensors required to monitor the temperature of a reservoir or air pollution in the environment is small. In contrast, for applications that require higher accuracy, such as military and disaster area monitoring, the number of sensors can be increased to thousands.

However, it is difficult to control the number of active sensors to a certain number. Due to the large number of sensors and the wide distribution area, it is not feasible to obtain information from all sensors for control. In addition, a large number of active sensors waste power and bandwidth, whereas few active sensors result in insufficient environmental information. In an actual network environment, owing to battery failure, sensor damage, and the addition of sensors, the total number of sensors changes frequently.

To address these issues, in [5], the authors first applied the Gur Game algorithm to control the number of active sensors and then defined the quality of service as the number of active sensors. In the Gur Game, each active sensor sends a packet to the base station. When the base station receives packets from the active sensors, it counts the total number of packets and calculates the probability of the sensor changing its state in the next round through the reward function. After several operations, the total number of active sensors gradually reaches the target value. In [6], the authors proposed the ACK algorithm, in which the base station individually sends feedback packets to each active sensor. Sleep sensors do not need to receive feedback packets, and the ACK algorithm uses a random-access mechanism to increase the lifetime of the entire network [6,7,8,9,10].

In this study, we propose a high reliability control algorithm with fast convergence and strong self-organization ability, called the sensor activity control algorithm (SACA). In SACA, there are two sub-processes: the adjustment procedure for the weight value of sensors and the transformation procedure for the state of sensors. SACA considers the relationship between the total number of active sensors and the target value and determines the state (ON or OFF state) of the sensors in the next round. Random access transmission technology is used between the sensor and the base station such that the sleep sensors do not need to receive the feedback packet from the base station, resulting in power savings.

### 1.2. Contributions

The three contributions of this research are as follows:(1)This paper proposes a sensor control algorithm with fast convergence. According to the application requirements, an algorithm that can provide fast convergence can avoid the waste of sensor power resources. SACA also uses random access technology, such that the sleep sensors do not need to receive the feedback packet from the base station and can truly achieve the purpose of dormancy and power saving. In the Gur Game algorithm, all sensors must receive feedback packets from the base station to switch their states.(2)The SACA algorithm is highly scalable and can be used in large-scale network environments. The sensor network is often composed of hundreds or thousands of sensor nodes. In the Gur Game, when the ratio of the target value to the total number of sensors is too large or too small, the number of active sensors cannot reach the target value.(3)SACA has a strong self-organization ability algorithm and can be applied in severe network environments, such as a network environment with a high death ratio (high damage ratio). Because sensors are usually deployed in dangerous areas, it is not possible to repair or add new sensors. Therefore, an algorithm with strong self-organization capabilities can enable sensors to operate in a severe network environment in real time, self-adjust, or maintain a specific target value.

The organization of the paper is the following. Section 2 describes the Gur Game and ACK methods, and Gur Game-based control algorithms for solving different problems. Section 3 describes preliminary assumptions of SACA. Section 4 describes the operation details of SACA, adjustment procedure for the weight value of sensors, and transform procedure of the state of sensors. Section 5 is the experimental analysis. We compare the differences among SACA, ACK, and Gur Game algorithms based on the algorithm’s convergence time, success rate, and self-organization ability of sensor nodes. In the last section, we present the conclusion and future work.

## 2. QoS Control Algorithms

### 2.1. Gur Game

In 2003, the Gur Game algorithm was first applied to control the number of active sensors [5]. In the Gur Game algorithm, it is assumed that the base station and sensor communicate in a signal hop and that they are time synchronized. Each sensor has a state machine. Before the Gur Game algorithm starts to operate, the sensor selects a state from the state machine as its initial state, and states 2 and 1 represent the active state, while states −2 and −1 are sleep states. As shown in Figure 1, the sensor must receive a data packet from the base station regardless of whether it is in an active or sleep state.

In each round, each active sensor sends a packet to the base station. The base station counts the total number of received packets and then computes the probability value *r*(*t*) of the state transition using the reward function, as shown in (1), where 0 ≤ *r*(*t*) ≤ 1. *Q*_1_ is the total number of packets received at time *t*, and *Q*_0_ is the target value. After the base station calculates *r*(*t*), it broadcasts the feedback packets to all sensor nodes. After receiving the feedback packets, the sensor randomly generates a random value *R* of 0–1. When the random value *R* is less than *r*(*t*), the sensors move to the 2 or −2 state of the state machine. Otherwise, the sensors move to the 1 or −1 state in the state machine. The sensor then determines the state of the next round based on the current position in the state machine. After several rounds of operation of the algorithm, the number of active sensors reaches (converges to) the target value.
*r*(*t*) = 0.2 + 0.8exp(−0.002(*Q*_1_ − *Q*_0_)^2^)(1)

Figure 2 shows the results corresponding to the reward function of the Gur Game approach, where the total number of sensors *m* is 100 and the target value *Q*_0_ is 35. After approximately 1200 epochs, the system achieved the target value. When the value of *r*(*t*) is close to one, the sensor is more likely to remain in its original state. After the Gur Game algorithm runs many times, the number of nodes in the operating state gradually reaches the target value, the calculated probability value *r*(*t*) approaches 1, and the state of the sensor nodes no longer changes.

### 2.2. ACK Algorithm

Another control method is the ACK algorithm [6]. Due to the limited power of the sensors, the state of the sensors does not change when the number of active sensors reaches the target value, which results in low power utilization efficiency. Using random access technology, the base station individually returns feedback packets to the active sensors. Therefore, sleep sensors do not need to receive feedback packets from the base station.

The ACK algorithm does not use a reward function to control the state of the sensor. Each sensor has a state machine, as shown in Figure 3. In the ACK algorithm, the state machine has three states, and each state contains the probability of a sensor being awakened (*P*_1_, *P*_2_, and *P*_3_). Sensors in the *P*_3_ state have a higher probability of being awakened than sensors in the *P*_2_ state, which have a higher probability than sensors in the *P*_1_ state (*P*_3_ > *P*_2_ > *P*_1_). The probabilities were 1.00, 0.10, and 0.05, respectively.

When the ACK algorithm starts to operate, each active sensor randomly selects a state as the initial state. Each sensor selects an active or sleep state based on the probability *P_i_* of its current state. The base station receives packets from active sensors, counts the number of packets received, and sends feedback packets to the active sensors individually. *Q*_1_ refers to the total number of packets received, and *Q*_0_ is the target value. The feedback packet contains the relationship between *Q*_0_ and *Q*_1_.

The base station, according to the relationship between the number of active sensors and the target value, indicates the sensors’ states to change. When the number of packets (total number of active sensors) is higher than the target value, the feedback packet indicates that the sensors should move to the left of the state machine. Conversely, if the number of packets is less than or equal to the target value, the feedback packet indicates that the sensor should move to the right side of the state machine. A round is complete when all active sensors receive the feedback packet. In the next round, the sensor decides between the ON and OFF states according to probability *P_i_*. After several runs, the number of active sensors gradually reaches the target value. Figure 4 shows the experimental results, where the total number of sensors *m* was set to 100 and the target value was set to 35. The experimental results show that the active sensors oscillate in the range of 34–42.

### 2.3. Related Works

Many studies have been conducted on the different aspects of QoS of sensor networks, such as power-saving [11,12,13,14,15,16,17,18,19], routing [20,21,22,23,24,25,26], throughput of networks [27], enhancement performance [28,29] and coverage [30,31,32], and. The main goal of this study was to autonomously control the number of active sensors. Considering that the sensor network may be in a harsh environment, a feasible approach must be able to reach the goal quickly, have strong robustness, and power-saving properties to cope with the harsh network environment.

There are two types of distributed control that use finite state automata to control the number of active sensors: (1) Gur Game-based and (2) random access-based algorithms. In the Gur Game, when the number of active sensors (QoS) reaches the target value, the sensor does not change its state, resulting in uneven power usage of the sensors. In [5], the network lifetime is defined as when the power of the first sensor node is exhausted, and the lifetime ends. In [6], the system begins to execute until the number of active sensors cannot maintain the target value and the network lifetime ends. In [11,12,13,14,15,16,17,18,19], the authors aim to solve the problem of sensor power imbalance and maximize the lifetime of the network system regardless of the definition. To solve the problem of the Gur Game in terms of power imbalance, the mechanism of using sensors to exchange states at a predetermined time can improve the network lifetime [12]. However, the sensor undergoes state exchange without considering the remaining power of the sensors and does not overcome the power imbalance problems. Therefore, in [13], the authors improved the state transition strategy in the finite state machine and modified the reward function for the relationship between QoS and the target value, aiming to speed up the convergence speed and make the power of the sensor even. In [14], the authors proposed a new adaptive method for QoS and energy management in IEEE 802.15.4 networks, called the “skip game”, which aims to maintain a trade-off between QoS and extending the lifetime of networks.

In [15], the authors proposed a QoS control algorithm called the Gureen Game. A Gureen Game divides the sensor state into active, standby, and sleep states. The standby state is the same as the Gur Game sleep state, whereas the sleep state turns off all of the functions of the sensor depending on whether the QoS is too high or too low. Adjust the reward probability of the situation. Each sensor uses a timer to change its state to achieve power-saving effects. Similar to the Gureen Game, in [16], the sensor state is divided into *T*_1_, *T*_2_, *R*, and S states, and the sensor in the *S* state will turn off all functions, thereby allowing the sensor to achieve true dormancy. The sensor in the *S* state will change to the *T*_1_ state at the desired time to achieve the purpose in terms of power balance.

Fast convergence can prevent the sensor from consuming unnecessary power resources and bandwidth. In [28], the authors proposed a fast convergence control method called QC^2^, which is mainly based on the relationship between the QoS and the target value, and using the virtual target shortens the convergence time. However, this method faces an environment in which the sensors continue to die. Due to continuous changes in QoS, the aim of power saving cannot be achieved.

In the multi-objective QoS control algorithm, the authors combined the two control objectives of QoS and coverage and proposed a compound control method that can control the coverage ratio in addition to QoS [30]. In [19,31,32], combining the two control objectives of QoS and node density, the control strategies for rectangular, circular, and various elliptical regions are given, and a dynamic adjustment mechanism based on the Gur Game algorithm is designed. In general, the convergence time of algorithms based on the Gur Game is slow and cannot be adapted to a large-scale network environment. When the QoS continues to change, owing to the slow convergence time, it will not be able to achieve the desired target value.

## 3. Preliminaries

Sensor power resources are limited, making energy supply in harsh environments difficult. In the Gur Game algorithm, the sink broadcasts feedback packets to all the sensors. Power resources are consumed when the sensors receive packets. Conversely, some studies show that the energy consumption of transmitting and receiving data are almost the same [5]. In the ACK algorithm, the sleep sensors do not need to transmit packets, and the waste of power resources is avoided by keeping the sleep sensor in a low power consumption state. Similar to the ACK method, in SACA, the sink transmits packets to each active sensor individually, and the sleep sensor is in a low energy state to avoid waste of power resources.

The sink determines whether the sensor is converted to the active state according to the change in the number of active sensors. Sensors are rewarded when they contribute to the outcome of the system and punished when they do not contribute to the system. Therefore, we use the target value and the number of active sensors to decide whether to reward or penalize the sensor nodes.

### 3.1. Sensor Activity Control Algorithm

This study proposes the SACA, which is a sensor control algorithm with high robustness, fast convergence, and power-saving capabilities. The sensor and base stations use random access technology to send and receive packets. The base station calculates the difference d between the total number of packets received at the current time (round) and the target value, and adds the difference *d* in the feedback packet, which is sent to the sensor. The sensor uses a weight value to determine whether it should switch its state. SACA mainly has two sub processors to control the number of active sensors, namely, the sensor adjustment procedure for the weight value of sensors and the transformation procedure for the state of sensors. In the state transition process, the active sensor determines whether it should continue in the active state or switch to the sleep state in the next round based on the difference recorded in the feedback packet. Sleep sensors do not need to receive feedback packets from the base station to achieve true dormancy. The sleep sensors gradually increase their weight value in each round to increase the probability of changing to the active state. In the adjustment procedure for the weight value of the sensors, the sensor decides whether to change the state or maintain the original state in the next round according to its weight value, *W_i_*.

Assume that *m* sensors and a base station are randomly deployed in the monitoring area. Before the system starts operating, the sensor randomly selects a state (e.g., an active or sleep state). Each sensor contained a weight value, *W_i_*, with an initial value of 0. The weight value, *W_i_*, affects the probability that the sensor changes its state in the next round. Each packet is transmitted from the sensor to the base station in one hop, and vice versa.

### 3.2. Operation Process of SACA

The following four steps comprise the entire operation process of SACA:

Step 1: each active sensor sends a packet to the base station. Sleep sensors do not need to send packets to the base station.

Step 2: the base station starts to count the total number of packets from the active sensors, determines whether the total number of active sensors (*Q*_1_) is close to the target value (*Q*_0_), and calculates the difference value (*d*) between *Q*_1_ and *Q*_0_. The difference value d is added to the feedback packet and sent to the active sensor.

Step 3 (adjustment procedure): the sensor executes the adjustment procedure for the weight values of the sensors. The active sensor determines whether to change its state (for example, maintain the original state or change from the ON state to the OFF state) according to the difference value *d* in the feedback packet. If the number of active sensors is too small (i.e., *Q*_1_ − *Q*_0_ < 0), set *W_i_* directly to 1, so that the sensors should remain in their original state. If the number of active sensors is too large in the current round (i.e., *Q*_1_ − *Q*_0_ > 0), set *W_i_* directly to 0, letting the sensor change to the sleep state. If the difference value d equals 0 (i.e., *Q*_1_ − *Q*_0_ = 0), the sensors change to the sleep state, and then the process goes to step 4.

Sleep sensors do not need to receive feedback packets from the base station. They increase their weight value, *W_i_*, in each round, as their main purpose is to turn on the active state to replace the active sensors. To increase the value of *W_i_*, we set an activation parameter(σ) and the sleep sensor increases its *W_i_*, and then the process goes to step 4.

Step 4 (transform procedure): after the sensors finish the adjustment procedure for their weight value, they execute the change-of-state procedure. The sensors randomly generate a probability value *r* and compare it with their own weight value *W_i_*. When *W_i_* > *r*, the sensor switches to the active state; otherwise, it changes to the sleep state. The probability value of *r* is between 0 and 1, and the larger the probability value of *r*, the greater the probability that the sensor is in a sleep state.

The four steps mentioned above, shown in Figure 5, represent one round. After the system executes several rounds, the total number of active sensors converges to the target value.

## 4. Methods

### 4.1. Adjustment Procedure for the Weight Value of Sensors

As previously mentioned, a sensor with a higher weight value, *W_i_*, will have a higher probability of maintaining the active state. Therefore, SACA uses the relationship between the total number of active sensors and the target value to adjust the weight of sensor *W_i_*. For the active sensors, the difference value *d* is used to determine whether the weight value *W_i_* should be increased or decreased (reward or punishment), where *d* is the difference between the target value and the total number of active sensors. The total number of active sensors is counted using the base station, which records the difference value in the feedback packet and sends it to the sensor according to the relationship between the total number of packets currently counted and the target value.

Only the active sensors receive feedback packets from the base station in the SACA. When the sensor is in the active state, the following three situations need to be considered:(1)When *d* > 0, there are too many sensors in the active state. Because the sensor node is currently in an active state, the sensor should change to a sleep state in the next round.(2)At *d* < 0, the total number of active sensors is currently insufficient. If the sensor is currently in an active state, it should be maintained in an active state in the next round.(3)When *d* = 0, the total number of active sensors reaches the target value. Therefore, the state of the sensor does not need to be changed. These situations are shown in Figure 6.

For the sensor in the sleep state, the weight value *W_i_* automatically increases every round. The main reason for this is that the sensor in the sleep state can replace the active sensors such that the active sensors can rest. The power loss between the sensors is then averaged. The smaller the weight value *W_i_*, the higher the probability of the sensor remaining in the sleep state or transitioning from the active state to the sleep state. Therefore, increasing the weight of the sensor increases the probability that the sensor is in an active state. The weight value *W_i_* is updated as shown in (2). The activation parameter of the weight value *W_i_* is σ, which affects the convergence time of the system. If the activation parameter is small, the weight value of the sensor needs more time to adjust, and the sensor needs to spend more time before it can be converted into an active state. In contrast, if the activation parameter is large, the probability of the sensor turning to the active state increases. The sensor in the sleep state can usually be used to supplement other sensors in the active state, for example, when a sensor in the active state is damaged or dead. Therefore, in a network environment with a high death ratio, setting the activation parameter is very important. In SACA, we set the activation parameter to 0.0001.

### 4.2. Transform Procedure of the State of Sensors

Each sensor was set with a weight value of *W_i_*, which was between 0 and 1. Before the system began to operate, the initial weight value of each sensor was set to 0. The weight value, *W_i_*, affects the state of the sensor in the next round. In the transformation procedure of the state of the sensors, the sensor generates a probability value *r* between 0 and 1. The state transition rules are described as follows:(1)When *r* greater than or equal *W_i_*, the sensor either remains in the sleep state or transitions from the active to the sleep state.(2)When *r* is less than *W_i_*, the sensor remains active or transitions from the sleep to active state. In other words, a sensor node with a higher weight value, *W_i_*, will have a higher probability of being in the active state.

We implemented the SACA and set the activation parameters to 0.0001. Figure 7 shows the execution result when the total number of sensors is 100 and the target value is set to 35.

We summarize the characteristics of ACK and SACA. SACA and ACK only receive packets from active sensors. The sleep sensors do not have to transmit data packets, and the sleep sensors are in a low power consumption state to avoid power waste. The difference between ACK and SACA is that the ACK algorithm uses a finite state machine with a fixed probability value as the probability that the sensor is in an active state. In SACA, each sensor is set with a weight value, which adjusts the probability of active and sleep modes according to the change in the number of active sensors and the target value.

## 5. Evaluations and Results

We designed three sets of experiments to verify the performance of SACA: convergence time, success ratio, large-scale network environment, and power loss. We observed that the active sensor quickly converged to the target value, avoiding the waste of power resources. In Section 5.2, we compare the convergence capabilities of the SACA, Gur Game, and ACK algorithms. We analyzed the performance of the three algorithms in terms of convergence time and success ratio to reach the target value. In Section 5.3, we apply the three algorithms to a severe network environment to verify self-organization ability. In Section 5.4, the power-saving capabilities of the three algorithms are compared. We also provide suggestions for improving SACA’s performance.

### 5.1. Performance Evaluation

Equation (2) shows the calculation algorithm for average convergence time. In *E* trials, the sum of the convergence times of successful convergence was averaged, where *c_j_* and *s* are the convergence times and number of successes with the target value set to *j*, respectively. Equation (3) is the calculation algorithm for the ratio of successful convergence, *S_i_*; that is, the percentage of successful convergence times in *E* trials. In the SACA, the activation parameters were set to 0.001 and 0.0001, respectively. Table 1 lists the experimental parameter settings.
(2)$$T_{i}=\frac{\sum_{j=0}^{m}c_{j}}{s},\;0\leq i\leq m$$
(3)$$S_{i}=\frac{s}{E}\times 100\%,\;0\leq i\leq m$$

### 5.2. Convergence Time and Successful Ratio Analysis

Figure 8 shows the execution result of the Gur Game; the target value was increased from 0 to 100. It can be observed from the experimental results that convergence to the target value occurred only when the target value was between 25 and 75. Therefore, when the target value is set too large or too small, the number of active sensors cannot successfully converge to the target value. In other words, the Gur Game has restrictions on its use.

Figure 9a,b shows the execution results of the ACK algorithm. When the probability of the *P*_1_ state of the state machine is set to 0.01, the success ratio can reach 100%, in which case the average convergence time does not exceed 250 rounds. The success ratio cannot reach 100 when *P*_1_ is set to 0 or 0.05. Therefore, in subsequent experiments, *P*_1_, *P*_2_, and *P*_3_ were set to 0.01, 0.1, and 1 for the ACK algorithm.

Figure 10a,b shows the results of SACA, where the target value was increased from 0 to 100. We set the activation parameters to 0.001 and 0.0001, respectively, and compared the activation parameters to the sensor convergence speed impact. According to the experimental results, when the activation parameter was set to 0.0001, the convergence success ratio of the SACA was close to 100%. The convergence time of SACA was shortened by approximately one-tenth that of Gur Game, which shows that SACA has strong self-organization and fast convergence speed ability.

A WSN usually consists of a large number of sensors. Therefore, when the number of sensors increases, the strong self-organization ability can avoid the waste of power resources and provide real-time environmental monitoring quality according to the requirements of the application. To further highlight the faster convergence speed of SACA compared to the ACK algorithm, we increased the total number of sensors from 1000 to 10,000. The target values were set at 30% and 70% of the total number of sensors, respectively. As shown in Figure 11, when the number of active sensors reached 3% of the target value, it indicated a successful convergence to the target value.

### 5.3. Self-Organization Ability Analysis with High Death Ratio

The actual network environment is often unpredictable. The sensor may be damaged or out of power, causing the total number of sensors to change. Therefore, we created a network environment with a high death ratio. We verified the SACA, Gur Game, and ACK algorithms in terms of self-organization and reconstruction capabilities. As in Section 5.2, in an environment that does not consider the death or damage of the sensor, when the activation parameter is small, fewer sensors change from the sleep state to the active state, which makes the system more stable. However, in a network environment with a high death ratio, the activation parameter is increased to reach the target value as soon as possible. To create a network environment with a high death ratio, we generated the life span of the sensor from an exponential distribution with an average value of 70. In addition, in this experiment, the activation parameter of SACA was set to 0.01.

Figure 12a shows the execution results of the SACA and ACK algorithms. The SACA algorithm took longer than the ACK algorithm to maintain the target value. The execution results of ACK and Gur Game are shown in Figure 12b, which barely stayed at the target value. The activation parameter affects the probability of the sensor transitioning from the sleep state to the active state. In other words, if the activation parameter is higher, the sensor has a higher probability of converting its state into an active state. In an environment that does not consider the death or damage of the sensor, when the activation parameter is small, fewer sensors will transition from the sleep state to the active state.

Therefore, in a network environment with small changes, we recommend setting smaller activation parameters to achieve a quick convergence to the target value. However, in a network environment with a high death ratio, owing to the large changes in the network environment, a larger activation parameter should be used to increase the speed of convergence to the target value. The results show that SACA has stronger self-organization and reconstruction capabilities than Gur Game.

### 5.4. Energy-Efficiency Analysis

Numerous studies define the network life cycle as follows: the system starts execution until the first sensor dies. The literature defines the network life cycle as follows: the system begins to execute until the number of active sensors cannot maintain the target value of 6. In many cases, we are concerned with the capability of the system to monitor environmental information. We adopted the latter definition in this study. The main purpose of this experiment was to illustrate the sink through individual transmission and broadcast transmission of QoS packets, as well as to analyze these two methods in terms of the effect of power consumption on network systems.

We will take the win nodes developed by the Rockwell Research Center as an example. Table 2 lists the various functions of the win nodes relative to the power consumption information. In this case, the sensor status can be divided into (a) transmission, (b) receiving, and (c) idle modes. In transitions between modes, which also exhibit varying power consumption, the power consumed by the conversion between the transmission mode and the receive mode is 19.35 mW, and that consumed by the conversion between the transmission mode and the idle mode is 14.75 mW [16,17]. Sensors can load battery types, such as lithium ions, NiMH, NiCd, and alkaline. In this experiment, 1100 mAh NiMH batteries were used to enable the network system to provide a total power of 1131 W. The total number of sensors *m* was set as 100, the target value *Q*_0_ was set at 35, and the evoke parameter of SACA was set to 0.001.

The execution results of the SACA and ACK approaches are shown in Figure 13a,b. The ACK approach is unstable compared with SACA in terms of maintaining the target value, which indicates that the number of active sensors is higher.

Although the Gur Game approach enables stable convergence for the target value, the number of active sensors cannot be maintained at the target value after 1000 rounds, as shown in Figure 14a.

Figure 14b shows the power consumption of the three approaches. The Gur Game approach is faster than the ACK and SACA approaches in terms of power consumption. The system power is exhausted by the Gur Game algorithm after approximately 2000 rounds. In the SACA and ACK approaches, the system power is exhausted at approximately 2700 rounds.

The initial power of each sensor is $E_{i}$ and the total number of sensors is *m*. Thus, Equation (4) can be used to calculate the total power of the entire network system as $E_{max}$. Equation (5) shows the ideal value $E_{G}$ of power consumption for the Gur Game approach, where $E_{T}$ is the power consumption of the sensor in the transmission mode, $E_{R}$ is the power consumption of the sensor in the receive mode, and *t* is the number of rounds. $E_{TR}$ is the power consumption of the conversion of sensors between the transmission and receive mode. Equation (6) gives the power loss ideal value $E_{A}$ of the SACA and ACK algorithms. $E_{TI}$ refers to the power consumption of the conversion of sensors between the transmission status and idle status, where $E_{G}$ and $E_{A}$ are not greater than $E_{max}$. Equations (5) and (6) illustrate that SACA is better than the Gur Game approach in terms of energy savings because sensors in sleep mode no longer receive any QoS packet from the sink. Moreover, the power consumption of the conversion of sensors between transmission status and idle status is less than that of the conversion of sensors between transmission status and receive status.
(4)$$E_{max}=\overset{m}{\underset{i=1}{\sum }}E_{i}$$
(5)$$E_{G}=\left[\left(Q_{0}E_{T}+\left(m-Q_{0}\right)E_{R}\right)+E_{TR}\right]t,\;\;\;E_{G}\;\leq \;E_{max}$$
(6)$$E_{A}=\left[\left(Q_{0}E_{T}+\left(m-Q_{0}\right)E_{I}\right)+E_{TI}\right]t,\;\;\;E_{A}\;\leq \;E_{max}$$

## 6. Conclusions

Effective control of the number of active sensors avoids wasting power and bandwidth resources. In this study, we propose a new method called the sensor activity control algorithm (SACA). The idea of SACA is to use the relationship between the number of active sensors and the target value to decide whether to change the state of sensors. SACA retains the random-access advantages of the ACK algorithm. In addition, SACA has more robust self-organization capabilities in the high death ratio scenarios of sensors and is better in terms of energy efficiency compared with the Gur Game and ACK algorithms.

In future work, we will consider the residual power of the sensor for adjusting the state of sensor nodes. Through a swarm intelligence algorithm or artificial intelligence technology, they record the relationship between the number of active sensors in the previous round and the target value to determine the state of the sensor in the next round. On the other hand, the transmission between nodes adopts a multi-hop mode, which is more in line with the real network environment. Therefore, we will consider the multi-hop network environment and delay issues.

## Figures and Tables

**Figure 1 sensors-22-02020-f001:**
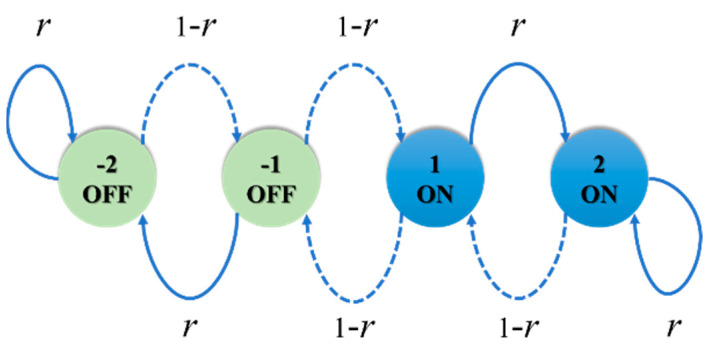
State machine used by the sensor in the Gur Game algorithm.

**Figure 2 sensors-22-02020-f002:**
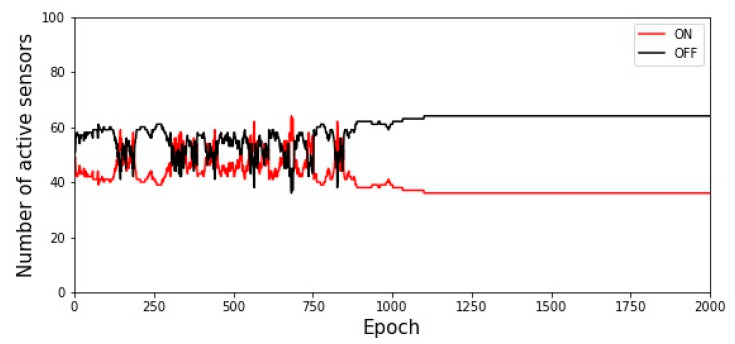
Execution results of the Gur Game approach when the target value *Q*_0_ is set as 35.

**Figure 3 sensors-22-02020-f003:**
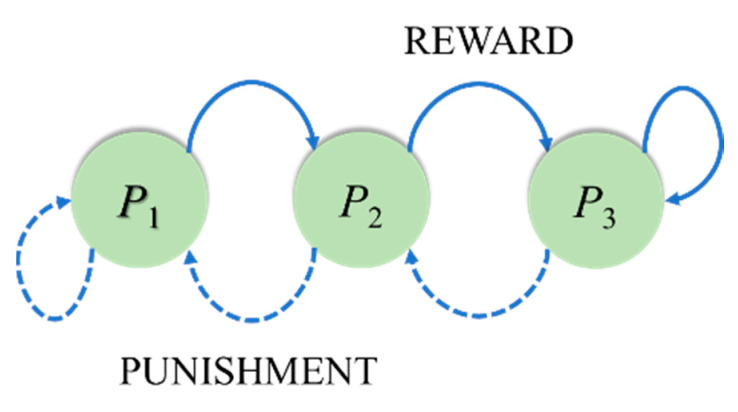
Finite-state machine in the ACK algorithm.

**Figure 4 sensors-22-02020-f004:**
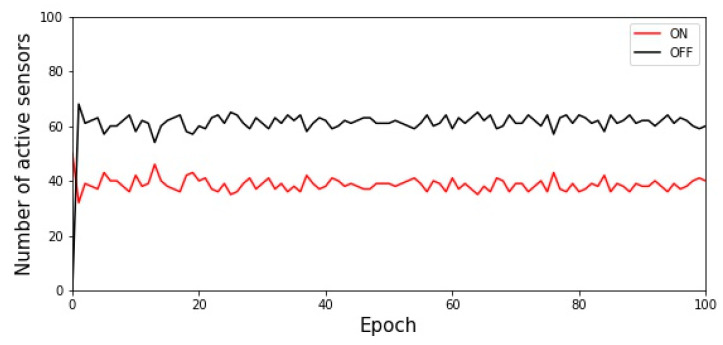
Execution result of the ACK approach when the target value is set as 35.

**Figure 5 sensors-22-02020-f005:**
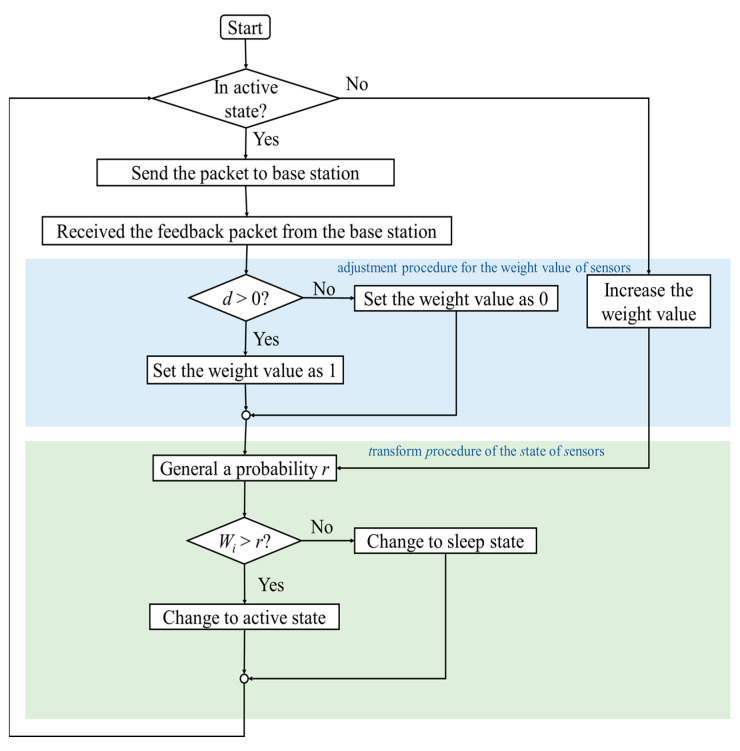
SACA operation flowchart, which includes the adjustment procedure for the weight value of sensors and transform procedure of the state of sensors.

**Figure 6 sensors-22-02020-f006:**
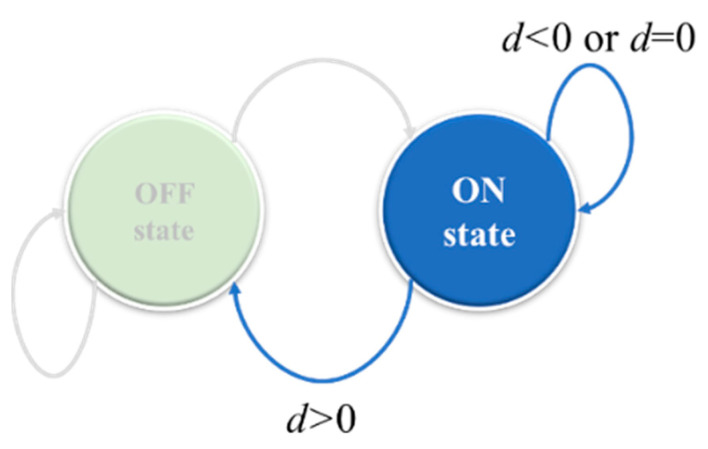
The active sensor performs a state transition according to the difference value *d* in the feedback packet.

**Figure 7 sensors-22-02020-f007:**
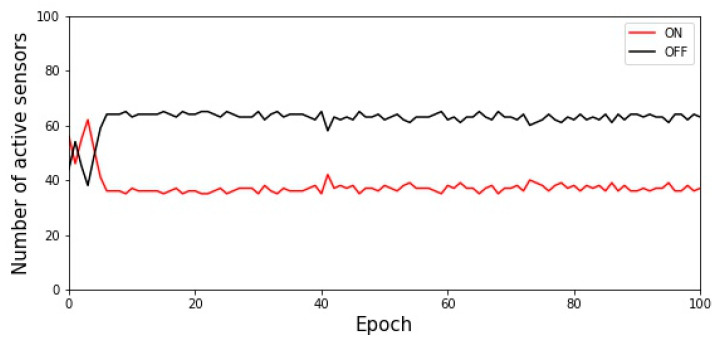
Execution result of the SACA when the target value is set as 35.

**Figure 8 sensors-22-02020-f008:**
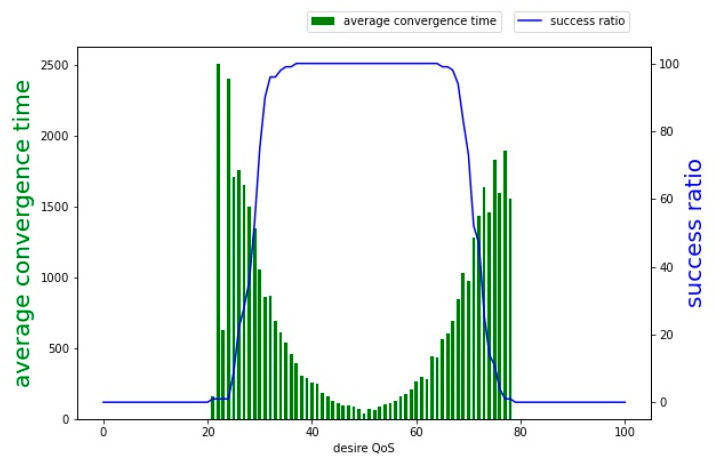
Average convergence time and success ratio in the Gur Game approach.

**Figure 9 sensors-22-02020-f009:**
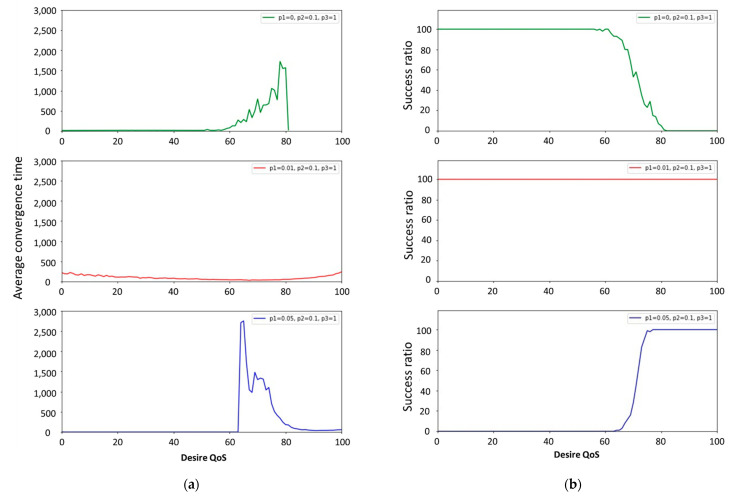
(**a**) shows the average convergence time; (**b**) the success ratio in the ACK algorithm.

**Figure 10 sensors-22-02020-f010:**
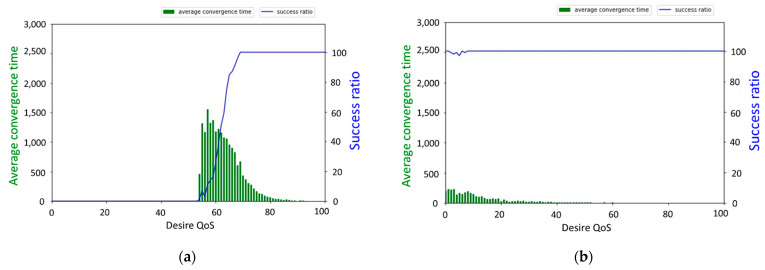
(**a**) shows the average convergence time and success ratio with σ is 0.001; (**b**) σ is 0.0001 in the SACA.

**Figure 11 sensors-22-02020-f011:**
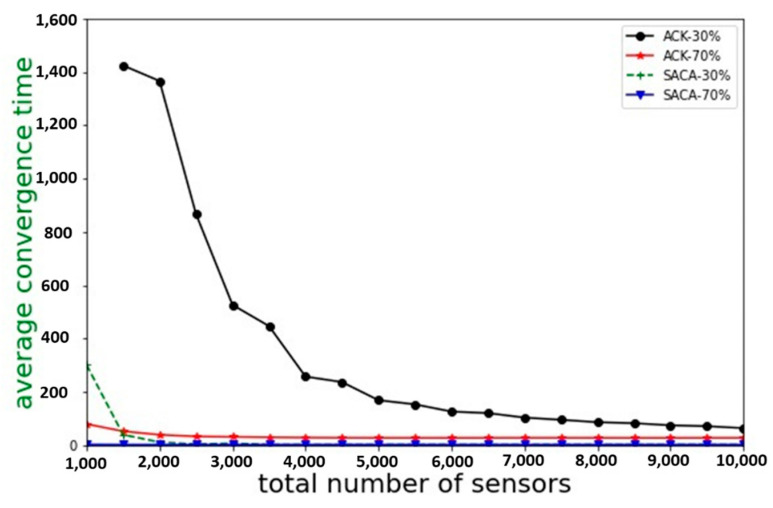
Comparison of the average convergence time of the ACK algorithm and SACA.

**Figure 12 sensors-22-02020-f012:**
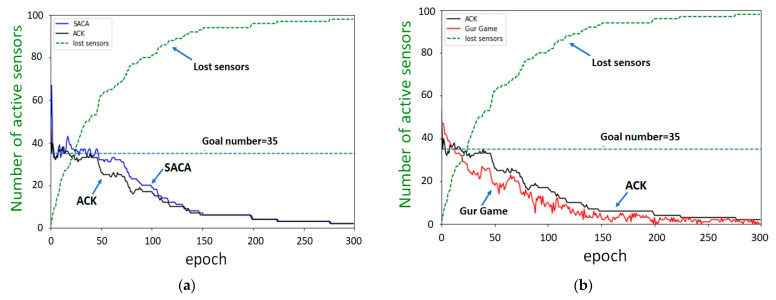
(**a**) SACA and ACK versus (**b**) ACK and Gur Game in highly death environments.

**Figure 13 sensors-22-02020-f013:**
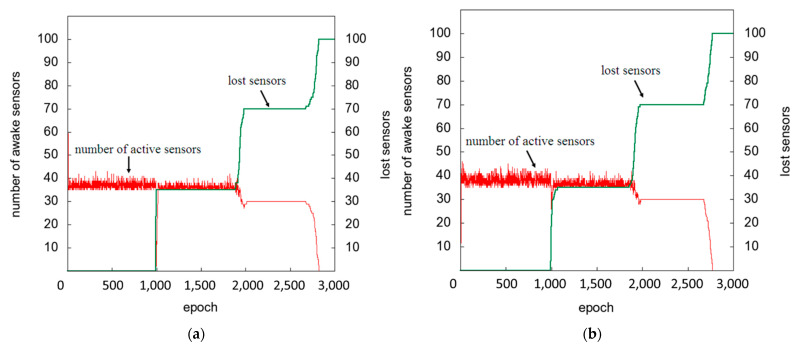
Variations of the number of active sensors for the (**a**) SACA, (**b**) ACK approach.

**Figure 14 sensors-22-02020-f014:**
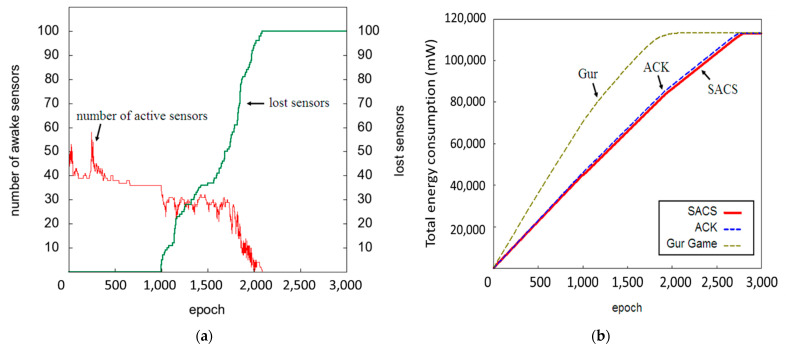
Variations of the number of active sensors for the (**a**) Gur Game approach, (**b**) the total power consumption of the network system for the SACA, ACK, and Gur Game.

**Table 1 sensors-22-02020-t001:** Experimental parameter settings.

Items	Parameter
Total number of sensors, *m*	10^2^, 10^3^–10^4^
Number of the sink	1
Target value, *Q*_0_	0–100 0.3, 0.7 (*Q*_0_/*m*)
Adjustment parameter, Δτ	1
Activation parameter, σ	0.001, 0.0001
Number of experiments	100
Execute rounds for each set of experiment	3000

**Table 2 sensors-22-02020-t002:** Power consumption of Rockwell’s wins nodes.

State Mode	MCU Mode	Sensor Mode	Radio Mode	Power (mW)
Transmission	Active	Active	Tx, Rx	1139.4
Receive	Active	OFF	Rx	409
Idle	OFF	OFF	OFF	40.7

## Data Availability

Not applicable.

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
