# Peer review of "A High-Robust Sensor Activity Control Algorithm for Wireless Sensor Networks"

_sensors, 2022, doi:10.3390/s22052020_

Round 1

Reviewer 1 Report

The authors proposed SACA - an algorithm for activity control of sensor nodes. The algorithm itself is simple, but the experiments proved that its efficiency is better than other ones studied. The problems/questions are as follows:

  1. There are some (mostly minor) uncertainties in the algorithms description, they are enumerated bellow.
  2. The difference in performance of SACA and ACK seems to be marginal (see fig. 12 or 13). Is it possible to improve this result by e.g. tuning algorithm parameters?
  3. Choosing parameter \sigma which is algorithm step seems to be delicate problem as large values will influence stability. Especially for so called 'high death scenario" larger value was used to boost convergence. Why this value was inappropriate for initial scenarios? It would be beneficial to implement step adaptation.

The most important problem is however different - its limited scope of proposed algorithm as it is only applicable to single hop networks with CA procedure. In such a network use of precomputed communication schedule would probably save more power thanks to contention avoidance. And such a schedule can be easily calculated (if feasible). On the other single hop network can cover limited space on the expense of high power consumption due to long range transmission. I encourage the authors to develop similar algorithm for multi hop network as I believe that it will allow more applications and will probably limit power consumption.

Detailed remarks:

1. The name SACA is misspelled several times.
2. line 204 - probably missinng "[]" around "5".
3. line 229 - does "adds" means that value of d is sent in feedback? Symbol "d" should be in italics.
4. p.3.2 and fig. 5: the role of r should be described better. Especially, is r bound to <0,1>?
5. p.3.2 and fig. 5: what is weight increment in fig. 5? Is it \sigma? It is mentioned in p.4 but it is too late and still unclear.
6. Formulas (2) and (3) should be described better. What is the difference/reason to compute them. What is "Suc".
7. Fig. 8 - how average convergence time and success ratio are computed? Is success ratio (3)?
8. Tab. 2 - transmission power seems to high, probably 1394mW? Please provide reference.

Reviewer 2 Report

The authors propose in this paper a new high reliability control algorithm with fast convergence and strong self-organization ability named Sensor Activity Control Algorithm (SACA). This algorithm is compared with two other related algorithms and the authors describe their contributions opposing to the drawbacks of the other mechanisms.

The paper is generally well written and clear.

The authors should provide recent related work to prove that this subject is still relevant. The most recent study in the references is another paper from the authors and previous to that only a short paper from 2019. Remaining works are already more than 5 years old.

Another limitation is in the results analysis section. Sub-sections 5.3 and 5.4 should be improved. I observe that the conclusions from the results are based on the comparisons of SACA with the Gur Game strategy because ACK and SACA are quite similar. Nevertheless, authors can further explain the pros of SACA against the ACK algorithm in the pointed situations where SACA has the advantage.

Lastly, future work should be addressed more thoroughly.
